# Assessing the Gel Quality and Storage Properties of *Hypophythalmalmichthys molitrix* Surimi Gel Prepared with Epigallocatechin Gallate Subject to Multiple Freeze-Thaw Cycles

**DOI:** 10.3390/foods11111612

**Published:** 2022-05-30

**Authors:** Zhihang Tian, Xin Jiang, Naiyong Xiao, Qiang Zhang, Wenzheng Shi, Quanyou Guo

**Affiliations:** 1College of Food Sciences & Technology, Shanghai Ocean University, Shanghai 201306, China; 17865561866@163.com (Z.T.); 13053523375@163.com (X.J.); xny931215@163.com (N.X.); qzzz1274@163.com (Q.Z.); 2National R & D Branch Center for Freshwater Aquatic Products Processing Technology (Shanghai), Shanghai 201306, China; 3East China Sea Fisheries Research Institute, Chinese Academy of Fishery Sciences, Shanghai 200090, China

**Keywords:** surimi gels, EGCG, gel quality, storage properties, freeze-thaw cycle

## Abstract

Epigallocatechin gallate (EGCG) with concentrations of 0–0.03% was added to manufacture surimi gels, respectively, while effects on gel quality and storage properties indicators during freeze-thaw (F-T) cycles were investigated. The results implied that the gel quality and storage properties of surimi gels added without EGCG were seriously destroyed during F-T cycles. The addition of EGCG could inhibit the decline of texture and gel strength. Moreover, EGCG has effect on inhibiting the microbial growth and the formation of off-odor compounds such as total volatile basic nitrogen (TVB-N) and malondialdehyde (MDA). Low-field nuclear magnetic resonance (LF-NMR) and water-holding capacity (WHC) results showed that immobilized water migrated to free water with the extension of F-T cycles. The scanning electron microscope (SEM) observed denser protein networks and smaller holes from the surimi gels added with EGCG. However, excessive (0.03%) EGCG showed the loose network structure and moisture loss. Overall, EGCG in 0.01–0.02% addition was good for resisting damage of surimi gels during F-T cycles.

## 1. Introduction

Surimi products, a kind of elastic and nutritious food with high protein and low fat, are made from frozen surimi by suwari and kamaboko. Frozen storage is a common way to limit protein oxidation and microbial growth, thus prolonging the shelf life of surimi products [1]. However, repeated freeze-thawing occurs inevitably in surimi products during processing, storage, transportation, sale, and consumption. Due to the temperature fluctuation, the gel quality of surimi products has been destructed by ice crystals, such as hardness reduction, moisture loss, and gel network fracture. The adverse biochemical reaction of surimi is occurred swiftly by ice crystals which leave the hole [2]. Besides, myofibrillar protein is the main part of surimi, which plays a key role in gel formation. With the increase of F-T cycles, the structural integrity of the surimi myofibrillar protein network gradually decreased by extrusion of ice crystals, thus affecting the texture, whiteness, and WHC of surimi products [3]. Microbial growth is also an important cause of quality changes in surimi products during frozen storage. How to effectively control the quality deterioration of surimi during freezing has always been a hot research topic. However, to our best knowledge, few studies on delaying quality deterioration of surimi products during freeze-thaw cycle.

Tea polyphenols, as the emerging food additive for aquatic products, have played a vital role in antioxidant and bacteriostatic properties [4]. The main component of tea polyphenols is catechins. There are four main catechins, including epicatechin (EC), epicatechin gallate (ECG), epigallocatechin (EGC), and EGCG [5]. Due to the ability of antiseptic and sterilization, EGCG was widely applied in the medical field to opposite cancer, virus, and obesity [6]. In the field of food processing, transglutaminase and EGCG had double cross-linking impact on myofibrillar proteins modification, and that EGCG could further improve the gel properties under the induction of transglutaminase [7]. The adducts reacted with EGCG and malondialdehyde could enhanced gel quality and fixed the microstructure of myofibrillar proteins [8]. Besides, EGCG plays an essential role in preserving the freshness of food. EGCG combined with squid ring shell oligosaccharides was used to extend the product life of yellowfin tuna fillets by 12 days at 4 °C [9]. Furthermore, the gum tragacanth-sodium alginate active coating containing EGCG and lysozyme could prolong the shelf life of large yellow croaker [10]. As a natural additive for use in frozen food, EGCG has been proved to be viable in aquatic product preservation. However, few studies are focused on the addition of EGCG to surimi gels.

The study aimed to sight how EGCG affected surimi products during F-T cycles. Aerobic plate count (APC), total volatile basic nitrogen (TVB-N), thiobarbituric acid reactive substance (TBARS), and pH were used to identify the state of corruption. Moreover, the texture properties, gel strength, whiteness, WHC, and moisture distribution were investigated to study the gel properties and storage properties of surimi gels. Collectively, the microstructure was analyzed the changes in the myofibrillar proteins network. This research can serve as an experimental foundation for the production and quality control of high-quality surimi products during frozen storage.

## 2. Materials and Methods

### 2.1. Materials and Reagents

Silver carp (*Hypophythalmalmichthys molitrix*) frozen surimi (AAA grade) was obtained from Jingli Aquatic Food Co., Ltd. (Honghu, China) and then cut into approximately 1 kg blocks. These surimi blocks were kept at −20 °C until further need. EGCG with a purity of 98% was provided from Shanghai Meryer Co., Ltd. (Shanghai, China). Other chemicals were of analytical grade and supplied from Sinopharm Chemical Reagent Co., Ltd. (Shanghai, China).

### 2.2. Preparation of Heat-Induced Gel and F-T Cycles

Raw surimi blocks were defrosted at 4 °C overnight until a semi-thawed state and cut into about 3 cm cubes. Then, the surimi cubes were blended using AM-CG108 food processor at 5000 rpm for 2 min. NaCl (2.5%) was added and the blending was continued for 2 min. Subsequently, different content EGCG (0%, 0.01%, 0.02%, and 0.03%) was sprinkled into the surimi and added ice water to adjust the moisture content of surimi to 78%, during which time the mixture was blended at 5000 rpm for 6 min. The entire procedure was kept at a temperature of less than 10 °C. Next, the surimi paste was filled into polyvinylidene chloride casings (25 mm diameter) to make surimi sausages (20 cm long) and make sure that both ends were sealed tightly to prevent water from immigrating. Finally, the two-step heating method was used, with 60 min at 40 °C followed by 30 min at 90 °C. After sterilization, the sausages were promptly chilled in ice water for 30 min before being frozen at −20 °C.

The F-T cycles were designed as follows. The one F-T cycle was defined that surimi gels were prepared by freezing at −20 °C for 72 h after thawing at 4 °C for 12 h. Repeated 6 cycles and selected 0, 2, 4, 6 cycles to measure relevant indicators.

### 2.3. Texture Profile Analysis (TPA)

Surimi gels were sliced into the cylinders with a height of 20 mm, and then equilibrated to room temperature for 30 min. Hardness, springiness, cohesiveness, and chewiness were estimated by TA-XT plus texture analyzer (Stable Micro Systems, Vienna, UK) equipped with probe P/50. The parameters were as follows: 3 mm/s pre-test speed, 1 mm/s test speed, 3 mm/s post-test speed, 50% compression strain, and 10 g trigger force.

### 2.4. Gel Strength Measurement

Gel samples were balanced at room temperature for about 30 min before testing. Breaking force (g) and deformation (cm) were measured in gel cylinders cut into 25 mm lengths, using TA-XT plus texture analyzer (Stable Micro Systems, Vienna, UK) equipped with a P/5S probe attachment. The gels were examined at a press down distance of 15 mm with a pretest speed of 2 mm/s, test speed of 1 mm/s, and post-test speed of 10 mm/s. Gel strength was calculated by the following Equation (1):(1)Gel strength (g·cm)=Breaking force(g)×Deformation(cm)

### 2.5. Color Measurement

The apparent color of the gel was measured by CR-400 colorimeter (Konica Minolta, Osaka, Japan) to analyze L* (lightness), a* (redness-greenness), and b* (yellowness-blueness). The whiteness (W) was computed in the following Equation (2):(2)$$W=100\;-\;{\left[{\left(100\;-\;L^{*}\right)}^{2}\;+\;{a^{*}}^{2}\;+\;{b^{*}}^{2}\right]}^{1/2}$$

### 2.6. Storage Properties Measurement

#### 2.6.1. Aerobic Plate Count (APC) Measurement

Samples were prepared using the approach of Xu et al. [11] with some modifications. Accurately 2 g of surimi gel was weighed, and 18 mL of normal saline (0.85%, *w*/*w*) was added. After aseptic homogenization for 3 min, 10-fold dilution was continued. Standard plate count agar was selected for APC. Sample liquid (1 mL) which needed 3 dilution gradients was injected into plate count agar medium and kept at 30 °C for 72 h. 2 parallels were set up per dilution gradient. Results of APC were expressed as log colony forming units (CFU)/g sample.

#### 2.6.2. Total Volatile Basic Nitrogen (TVB-N) Measurement

The TVB-N value was determined according to Wang et al. [12] using FOSS 8400 Kjeldahl Nitrogen apparatus (FOSS Corporation, Denmark) with slight modification. The samples of surimi gels (10.0 g) were minced and mixed with 10.0 g magnesium oxide slightly in a digestive tube. Following distillation, the volatile nitrogen was recovered and titrated with 0.1 M hydrochloric acid in a 1% boric acid solution (*w*/*v*).

#### 2.6.3. pH Value Measurement

The surimi gel was equilibrated at normal temperature for 30 min. Then, 2 g of gel sample was homogenized with 18 mL of deionized water for 1 min by AD200L-P high-speed dispersing homogenizer. The homogenate was centrifuged at 10,000 r/min for 10 min, followed by filtering and retaining the supernatant. The pH was measured with an FE28 pH meter and performed at least in triplicate.

#### 2.6.4. TBA Measurement

First, 1 g of sample was weighed and added to 10 mL of 7.5% trichloroacetic acid (containing 0.1% of EDTA). After being homogenized for 1 min at 15,000 rpm, samples were left in the ice bath for 10 min and then filtered. To 3 mL of filtrate in test tubes, 3 mL of 0.02 mol/L 2-thiobarbituric acid in distilled water was added. The sample solution was reacted at boiling bath for 40 min. Finally, the absorbance was measured at 532 nm after cooling with ice water. A standard curve was built by 1,1,3,3-Tetraethoxypropane (TEP).

### 2.7. Water-Holding Capacity (WHC) Measurement

Surimi gels were sliced into 5 mm cylindrical slices and weighed as W_1_, which closed to 3 g. Next, the gel sample was wrapped by a piece of double-layer filter, and centrifuged at 5000 r for 15 min. Finally, the filter paper was removed and the weight of samples was measured as W_2_. The Equations (3) and (4) were used to determine WHC.
(3)$$\begin{array}{c} \mathrm{Centrifugal}\;\mathrm{loss}\;\left(\%\right)=\frac{W_{2}}{W_{1}}\;\times \;100 \end{array}$$
(4)$$\begin{array}{c} \mathrm{WHC}\;\left(\%\right)=\frac{G-\mathrm{CL}}{G}\times 100 \end{array}$$

Thereinto, G represented the moisture content determined by drying the sample at 105 °C until constant weight and CL was the centrifugal loss.

### 2.8. Low-Field Nuclear Magnetic Resonance (LF-NMR) Spin-Spin Relaxation (T_2_) Measurement

The moisture distribution and relaxation time were determined using the approach of Jiang et al. [13] by a Niumag Pulsed NMR analyzer (MesoMR23-060H-I, Niumag Electric Co., Shanghai, China). Surimi gels were cut into cylinders (20 mm × 20 mm), and transverse relaxation (T_2_) was determined using the Carr-Purcell-Meiboom-Gill (CPMG) pause sequence. The data from 8000 echoes were collected throughout the course of 8 scan repeats. All measurements were performed at 32 °C in triplicate.

### 2.9. Magnetic Resonance Imaging (MRI) Measurement

The measurement of MRI was described by Cheng et al. [14] with slight modifications. Samples were wrapped in plastic wrap and put into an MRI tube. Proton density imaging was obtained by MSE imaging sequence with the main following parameters: Repetition Time = 500 ms, Echo Time = 18.2 ms, and Average = 6. Finally, the condition of mapping and pseudocolor was provided by Shanghai Niumag Electric Co., Ltd.

### 2.10. Microstructure Measurement

The samples were sliced into 3 mm × 3 mm × 1 mm pieces by surgical knife and fixed with glutaraldehyde (2.5%, *v*/*v*) for 12 h at 4 °C. Then, 0.1 M phosphoric acid buffer (pH 7.2–7.4) was used to rinse the gel samples three times. Afterwards, a serious density of ethanol (30%, 50%, 70%, 80%, 90%, and 100%) was added in gel samples for gradient elution. The dehydrated samples were immersed in a mixture of ethanol and tert butanol (3:1, 1:1, 1:3, 0:1) to remove the ethanol. The processed pieces that sputter-coated with gold after freeze-drying were observed by a Scanning Electron Microscope instrument (Hitachi SU5000, Hitachi High-Tech Co., Ltd., Shanghai, China) at an acceleration voltage of 5 kV.

### 2.11. Statistical Analysis

Each experiment was repeated three times. Statistical Package for Social Science 26.0 software (SPSS Inc., Chicago, IL, USA) was used to analyze the experimental data, and the mean ± standard deviation (SD) was provided. Duncan’s multiple range test method was used to compare significant differences using one-way analysis of variance (ANOVA) at the 0.05 level.

## 3. Results and Discussion

### 3.1. TPA

TPA is widely applied to research the physical characteristics of surimi gels, which become an effective method to investigate texture [15]. The change in hardness, springiness, cohesiveness, and chewiness of gel matrix with different content EGCG during F-T cycles were shown in Figure 1. Samples were compressed twice to provide texture test curve and grasp the characteristic parameters of the gel matrix [16]. Texture-related properties were overall decreased with increasing F-T cycles. The phenomenon might be attributed to the disruption of the F-T cycles to the formation, melting, and regeneration of ice crystals, resulting in larger and more irregular ice crystals that physically disrupt the protein network than ice crystals before F-T cycles [17]. Besides, the growth of ice crystals affected the tightness of the myofibrillar network and part of the water was migrated outside the gel, which might be associated with the gel strength of surimi gel [18]. Thereinto, Figure 1a,d showed that F-T cycles had a substantial impact on hardness and chewiness of the 0% EGCG group. The hardness and chewiness changed the most between the zeroth and second F-T cycles, demonstrating that the quality of surimi gel was seriously degraded during this period. Besides, there were significant differences in chewiness of the sample with 0.01% EGCG and three other samples after sixth F-T cycles (*p* < 0.05). Figure 1b,c revealed that the springiness and cohesiveness reduced considerably after fourth F-T cycle (*p* < 0.05). It was indicating that negative impact of the springiness and cohesiveness was mainly occurred in the late F-T period. Moreover, the 0.01% EGCG group had better springiness and cohesiveness, illustrating the effective preservation of texture. Moreover, the downward trend of hardness, springiness, cohesiveness, and chewiness of gels without EGCG was larger than those in the other added amount groups during the F-T cycles. These findings showed that EGCG had a beneficial influence on surimi gel F-T stability.

### 3.2. Gel Strength

Figure 2 showed that breaking force, deformation, and gel strength of samples were all decreased due to the F-T damage. Generally, the breaking force could characterize the hardness of gels and the deformation could represent the gel flexibility and elasticity [19]. Correspondingly, breaking force of samples rose followed by a reduction as the EGCG content increased after each F-T cycle, and the gels added with 0.01% EGCG had the maximum breaking force. The phenomenon was consistent with the result of TPA. As the main functional protein, the higher-order structure of myosin unfolds after heating to form a stable network structure, which is intertwined with each other through hydrogen bonding to form fibrous macromolecules [18]. It was hypothesized that the gel strength treated with EGCG was connected to the changes in protein structure of surimi gels. Li et al. [7] reported that EGCG could promote the cross-linking of myofibrillar proteins and increase the gel strength of surimi gels with the non-covalent interactions such as hydrophobic interaction and hydrogen bond. However, the group which treated high content of EGCG exhibited poor gel strength in Figure 2c. A possible explanation for this might be the self-aggregation of EGCG, which caused the loss in the ability of myofibrillar proteins cross-linking [20].

### 3.3. Color

Whiteness is the most intuitive indication for consumers to identify surimi quality. L*, a*, b* values, and whiteness of surimi gels, with and without EGCG, as affected by F-T cycles were shown in Table 1. As the cycles of F-T increased, the lightness decreased significantly (*p* < 0.05). The more the F-T cycle is, the worse quality the gels exhibit, the dimmer the lightness of the gel surface is. Compared with the 0% EGCG group, the group treated with EGCG exhibited low whiteness. There was a view that the tight gel network has positive effect on light absorption, which caused the deep color of gels [21]. Combined with the previous analysis, the greater the gel strength is, the lower the whiteness of surimi gel is. Besides, the quinones obtained by EGCG oxidation were brown, which resulted in the decline of whiteness. The redness-greenness of aquatic products might be related to the oxidation of myoglobin and the reduction of methemoglobin [22]. In the whole F-T cycles, the a* value of each group had no significant change. It was supposed that the myoglobin was removed when the surimi was rinsed, resulting in the stability of a* value. In addition, the b* value of each EGCG group was significantly lower than the control group. It was reported that a small amount of fat oxidation and pigment degradation in surimi could affect the color of surimi gels, especially the yellowness-blueness [23]. It might be that EGCG could better inhibit fat oxidation.

### 3.4. Storage Properties

#### 3.4.1. Aerobic Plate Count (APC)

One of the major contributors to the spoiling of surimi products during storage is microbial development, which has an impact on shelf life [24]. Throughout the F-T cycles, the APC of surimi gels showed an upward trend, especially from the zeroth to second F-T cycles. It might be that fat oxidation was accelerated, which obtained foul-smelling rancid products due to the F-T cycles. Therefore, the conditions for microbial contamination were created.

As the EGCG content increased, the APC of surimi gels decreased after each F-T cycle, confirming the antibacterial effect of EGCG. Some assumptions could be explained to describe the results of APC: (1) EGCG destroyed the structural integrity of individual microorganisms, resulting in the release of intracellular components and functional impairment. (2) Respiration of microorganisms was influenced. (3) Proteins and enzymes of microorganisms were destroyed. (4) EGCG disturbed the metabolism of microorganisms with chelating metal ions [25]. EGCG has strong antioxidant properties [26]. Klancnik et al. [27] discovered that the growth of *L. monocytogenes*, *E. coli*, and *C. jejuni* was prevented by the role of AlpE (ethanolic extract of *A. katsumadai* seeds) and EGCG in jointly protecting the surimi gels. EGCG has been shown to have regulatory activity and strong antibacterial activity against Gram-positive and Gram-negative bacteria [28].

#### 3.4.2. Total Volatile Basic Nitrogen (TVB-N)

TVB-N is another important indicator for evaluating the storage properties of surimi gels [29]. In the process of corruption, volatile ammonia, and biogenic amines were obtained from proteins, amino acids, and nitrogen-containing compounds, which decomposed by microorganisms [30]. As can be seen from Figure 3b, the TVB-N values of the control group, 0.01%, 0.02%, and 0.03% EGCG group in the zeroth F-T cycle were respectively 2.77, 2.50, 2.61, and 2.61 mg/100 g. Subsequently, the value of the control group increased rapidly and reached 7.69 mg/100 g on the sixth F-T cycle, which was higher than other experimental groups. It was illustrated that EGCG treatment helped to suppress the development of TVB-N and remain the storage properties of surimi gels. Correspondingly, the curve trend of TVB-N value was similar to the APC.

#### 3.4.3. pH Value

The changes in pH of surimi gels with different EGCG content during F-T cycles were shown in Figure 3c. With the increase in the times of F-T cycle, the pH value of surimi gels was first increased, and then decreased. Various basic nitrogenous compounds such as amines and trimethylamine were generated due to the decomposition of protein, which induced the increase in pH [31]. During 0–2 F-T cycles, the pH of the control group rose faster than other groups, indicating that the protein decomposition of the control group was more serious. The changes in pH correspond to previous TVB-N results. Thereafter, depending on more suitable growth conditions supplied by F-T cycles, microorganisms decomposed small organic molecules to generate acidic substances [32].

#### 3.4.4. TBA

TBA value reflects the degree to which lipids in surimi products are oxidized to malonaldehyde [33]. The occurrence of lipid deterioration is inevitable in F-T meat [34]. It could be seen in Figure 3d that the TBA value of surimi gels increased continuously during the F-T cycles. After the fourth F-T cycle, the TBA value of the control group increased rapidly and was significantly higher than EGCG group (*p* < 0.05). In comparison to the control group, the TBA value of surimi gels was dramatically lowered when EGCG was added. The reason was that EGCG contains eight phenolic hydroxyl groups, which possess the ability to scavenge free radical and slow down or terminate free radicals chain reactions effectively as hydrogen donors [35,36].

### 3.5. Water-Holding Capacity (WHC)

WHC, defined as the ability to retain water in samples, is reflected by centrifugal loss after external influence [37]. The changes in WHC of surimi gels treated with different EGCG content during F-T cycles were shown in Figure 4. As the F-T cycles increased, the WHC of each group decreased. During freezing, the protein network was disrupted by growing ice crystals, leading to tissue deformation and moisture loss [38]. Therefore, WHC is directly affected by the density of the gel network. When the F-T cycles were not started, there were substantial variations between the samples with and without EGCG. Between the fourth and sixth F-T cycles, the WHC of surimi gel added without EGCG declined seriously from 86.06% to 83.34%. Among the samples added with EGCG, the 0.02% group maintained a high WHC in the whole F-T cycle. It represented that EGCG could effectively suppress the decline of WHC. However, the WHC of 0.03% EGCG group was significantly lower than the group added with 0.01% and 0.02% EGCG after the fourth F-T cycle (*p* < 0.05). It was assumed that a high dose of EGCG led to excessive crosslink of myofibrillar protein that induced negative aggregation. The loose gel network structure was obtained due to the decreasing uniformity and density. It resulted that surimi gels could not effectively intercept water.

### 3.6. Moisture Mobility and Distribution Analysis

#### 3.6.1. Moisture Mobility

Water mobility and distribution in the gel matrix may be explored using LF-NMR, which is a fast and non-destructive approach [39]. T_2_ represents the degree of water freedom. According to Figure 5a, three kinds of peaks appeared at approximately 0.1~10 ms (T_2b_), 20~200 ms (T_21_), and 300~1500 ms (T_22_), which represented bound water, immobilized water, and free water, respectively.

The T_21_ and T_22_ relaxation periods of the surimi gels were dramatically lengthened when the F-T cycles were extended. Due to the crystallization and recrystallization of water, the moisture migrated outside the protein networks and the degree of water freedom inside the gel was changed [40]. The shorter the relaxation time is, the higher the binding capability between protein molecule and water is, and the stronger the water stability is. The relaxation time of the other groups differs from that of the control group after the addition of EGCG. Thereinto, surimi gels treated with 0.02% EGCG exhibited shorter T_21_ and T_22_. In addition, the peak corresponding to the relaxation time T_21_ and T_22_ of 0.02% EGCG group shifted to the left in each F-T cycle and it was found that the value of the second peak decreased remarkably. It implied that EGCG can inhibit the fluidity of moisture and control the loss of water, which was in accordance with the WHC.

The percentage of peak area (P_T2b_, P_T21_, and P_T22_) can objectively reflect the relative moisture content of surimi gels in Figure 5b. EGCG had no influence on the alterations in PT_2b_ of the samples. It might be that the bound water was closely linked with proteins and had low fluidity during the F-T cycles. Furthermore, F-T cycles caused P_T21_ to decrease, while P_T22_ increased. When thawing, the internal moisture migrated to large network gaps. After freezing, bigger ice crystals were formed to extrude the protein network, resulting in the reduction of reabsorbed water while thawing again. There has been researching pointed out that myofibrillar protein denaturation reduces protein binding capacity for water during the F-T cycles, thus inducing the conversion of immobilized water to free water [41]. Similar observations were also reported in previous studies [42]. In the fourth F-T cycle, the P_T22_ of 0.02% EGCG group was 6.08%, which was lower than the control group (7.62%), indicating that EGCG could effectively inhibit the growth of free water. The result of the sixth F-T cycle was also in agreement with above phenomenon. EGCG hastened the formation of protein gel and increased the degree of cross-linking of myofibrillar proteins. Therefore, dense protein networks could limit the conversion of immobilized water to free water. However, a too dense protein network caused by excessive cross-linking (0.03% EGCG) could not maintain the state of moisture, which was similar to the result of WHC.

#### 3.6.2. Moisture Distribution

MRI is also an auxiliary way to evaluate moisture distribution in surimi gels during F-T cycles [43]. Pseudo color image (value range: 0~255) produced by nuclear magnetic resonance could directly observe the situation of moisture content and distribution in Figure 6. The more the hydrogen protons exist, the redder the image exhibits, and the higher water content the gel has. On the contrary, the blue area represents low moisture. During the F-T cycles, the signal strength decreased gradually, illustrating that the partial moisture of the surimi gels was lost. Gels added with 0.01% and 0.02% showed stronger signals, which implied that they had higher water content, which supported the previous results. Li et al. [7] reported that the fraction of immobilized water rose with the addition of EGCG and transglutaminase. Overall, the dense gel network induced by EGCG is the primary cause for preserving moisture and making the water distribution uniform.

### 3.7. Microstructure

The changes in microstructure inside the gel system observed by scanning electron microscope can intuitively explain the close relationship between 3D network structure and gel properties [44]. Figure 7 showed the pore size and compactness of the protein network structure at 10,000× magnification. After two F-T cycles, large holes were obviously observed in the control group and there were traces of breakage in the protein network. However, samples treated with EGCG presented less ice crystal damage and better protein structure. In the initial stage of F-T cycle, 0.01% and 0.02% EGCG group have already shown a dense and regular gel network. Changes in microstructural compactness may account for the reduced whiteness of surimi gels with EGCG. It was reported that the addition of EGCG could expand myofibrillar protein structure and expose reactive groups in proteins, inducing massive protein aggregation relies on chemical bond action [45]. Furthermore, the declined trend of gel strength is related to the integral and dense extent of the gel microstructure, which is consistent with previously reported results [2].

## 4. Conclusions

Changes in the gel properties and storage properties of surimi gels treated with four different dosages of EGCG during F-T cycles were investigated. The addition of EGCG could delay the decrease of hardness, springiness, chewiness, and gel strength. By the antioxidant and bacteriostatic properties of EGCG, the indicators of storage properties such as APC, TVB-N, pH, and TBA were also significantly inhibited. Meanwhile, the higher the EGCG content is, the greater the inhibitory effect is. In addition, LF-NMR and WHC indicated that immobilized water migrated to free water, leading to severe water loss during F-T cycles. Compared with the control group, 0.02% EGCG group could better maintain the moisture of gels. This is attributed to the cross-linking of myofibrillar proteins induced by EGCG, which enhances the binding ability of the gel to water. Through the microstructure, EGCG groups represented denser protein networks and smaller holes. Namely, samples treated with EGCG suffered less damage during F-T cycles. However, overdose (0.03%) of EGCG exhibited fragile and loose network structure, which caused poor gel strength and WHC. In conclusion, this study found that 0.01–0.02% EGCG is an ideal added amount for surimi gels, which provides a data basis for resisting damage during F-T cycles.

## Figures and Tables

**Figure 1 foods-11-01612-f001:**
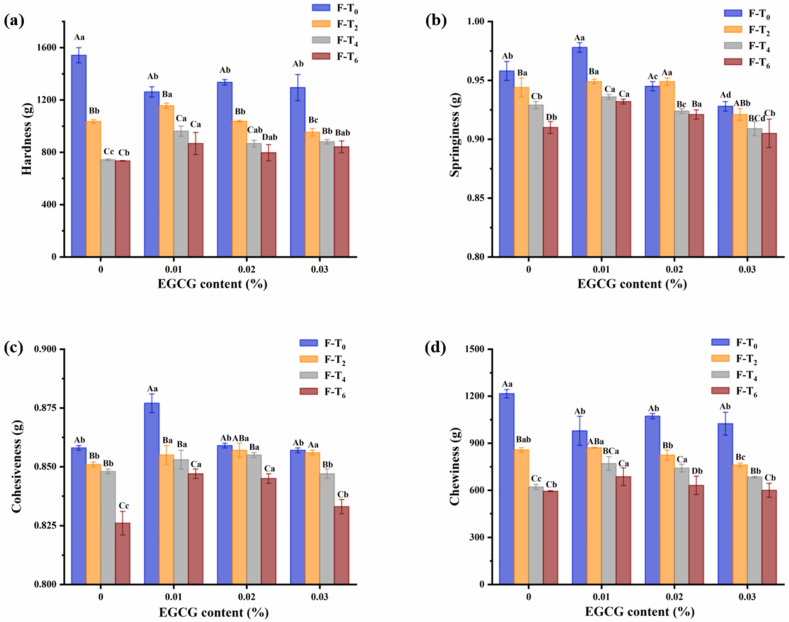
Changes in hardness (**a**), springiness (**b**), cohesiveness (**c**), and chewiness (**d**) of surimi gels treated with different content EGCG during F-T cycles. Uppercase letters indicate significant difference (*p* < 0.05) between different F-T cycle, lowercase letters indicate the difference between gels with different EGCG content (*p* < 0.05). F-T_0_: the zeroth freeze-thaw cycle; F-T_2_: the second freeze-thaw cycle; F-T_4_: the fourth freeze-thaw cycle; F-T_6_: the sixth freeze-thaw cycle.

**Figure 2 foods-11-01612-f002:**
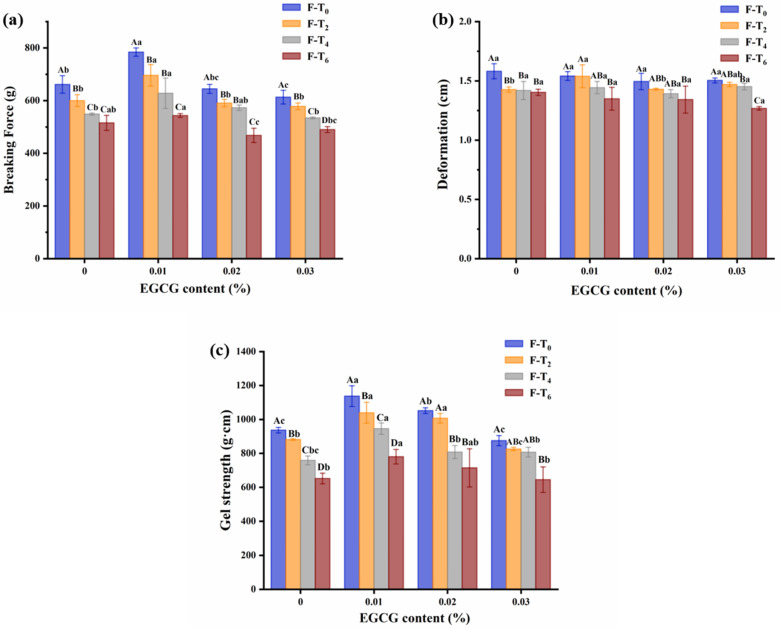
Changes in breaking force (**a**), deformation (**b**), and gel strength (**c**) of surimi gels treated with different content EGCG during F-T cycles. Uppercase letters indicate significant difference (*p* < 0.05) between different F-T cycle, lowercase letters indicate the difference between gels with different EGCG content (*p* < 0.05). F-T_0_: the zeroth freeze-thaw cycle; F-T_2_: the second freeze-thaw cycle; F-T_4_: the fourth freeze-thaw cycle; F-T_6_: the sixth freeze-thaw cycle.

**Figure 3 foods-11-01612-f003:**
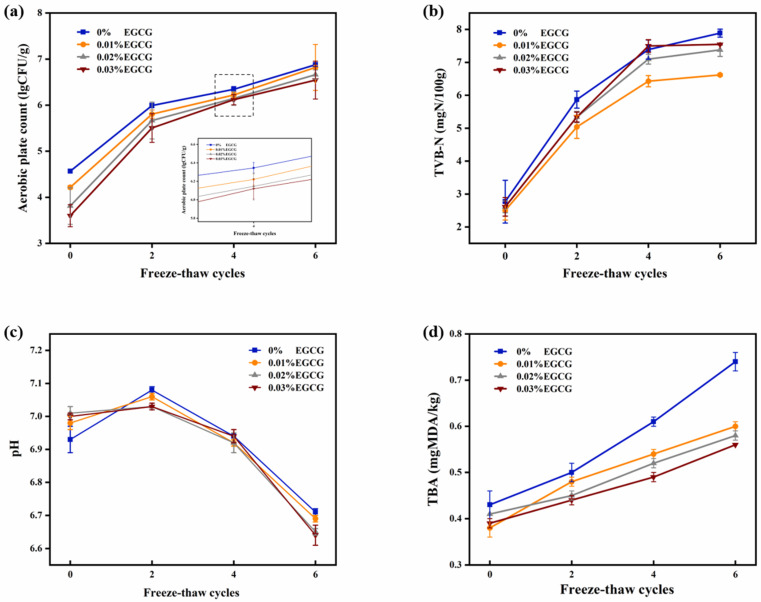
Changes in APC (**a**), TVB-N (**b**), pH (**c**), and TBA (**d**) of surimi gels treated with different content EGCG during F-T cycles.

**Figure 4 foods-11-01612-f004:**
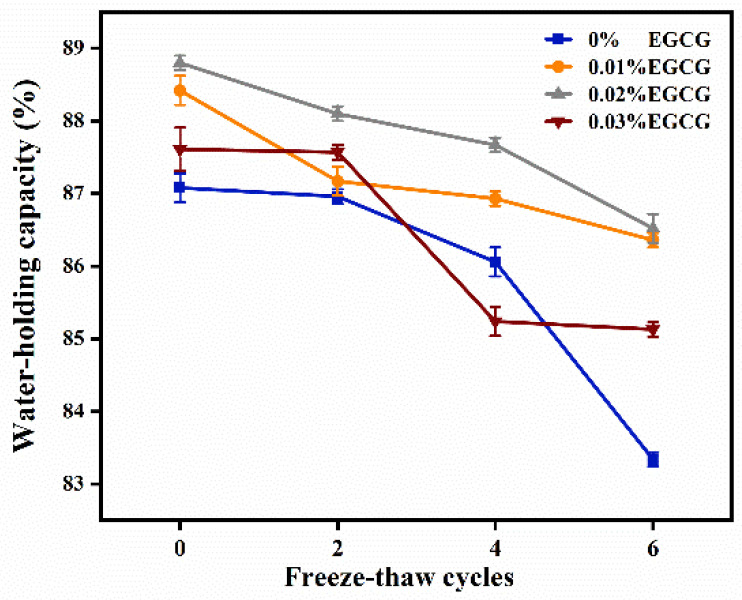
Changes in WHC of surimi gels treated with different content EGCG during F-T cycles.

**Figure 5 foods-11-01612-f005:**
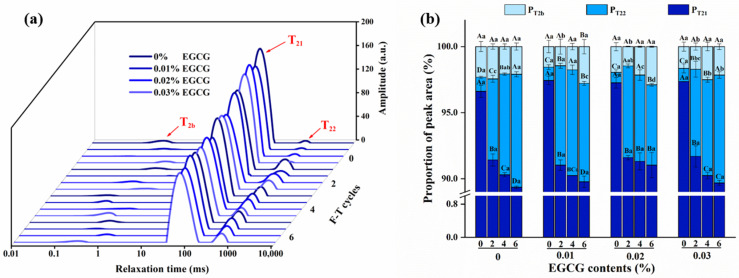
Changes in T_2_ relaxation times (**a**) and peak area proportion (P_T2b,_ P_T21,_ P_T22_) (**b**) of surimi gels treated with different content EGCG during F-T cycles. Uppercase letters indicate significant difference (*p* < 0.05) between different F-T cycle, lowercase letters indicate the difference between gels with different EGCG content (*p* < 0.05).

**Figure 6 foods-11-01612-f006:**
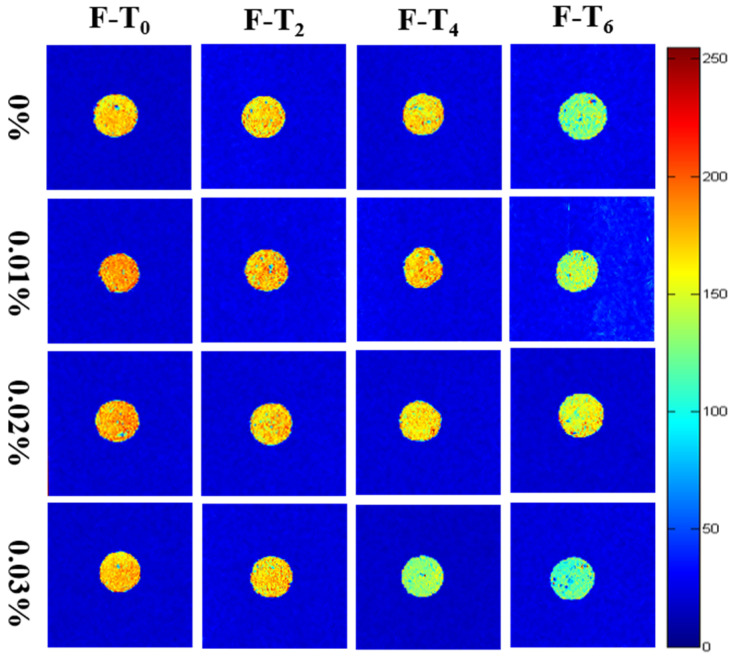
Changes in NMR pseudo color images of surimi gels treated with different content EGCG during F-T cycles. 0%, 0.01%, 0.02%, and 0.03% represented 0%, 0.01%, 0.02%, and 0.03% EGCG content, respectively. Caption: see Figure 1.

**Figure 7 foods-11-01612-f007:**
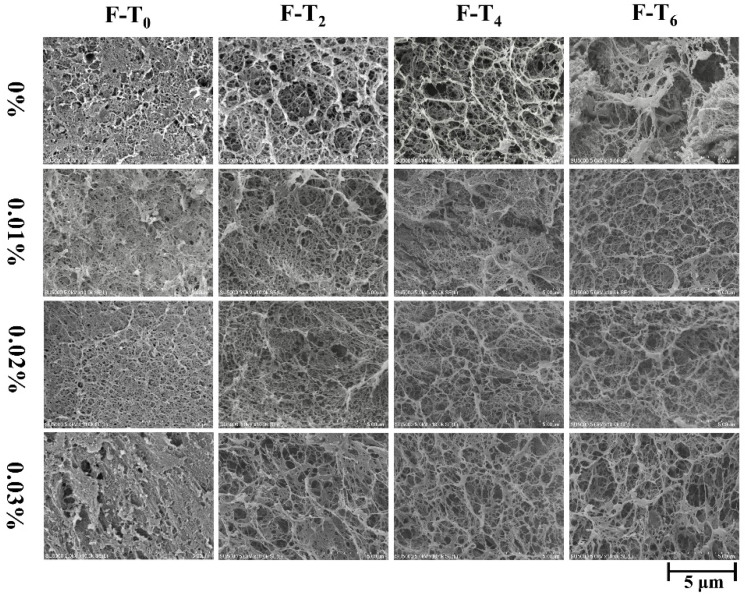
Changes in microstructure micrographs (magnification ×10,000) of surimi gels treated with different content EGCG during F-T cycles. 0%, 0.01%, 0.02%, and 0.03% represented 0%, 0.01%, 0.02%, and 0.03% EGCG content, respectively. Caption: see Figure 1.

**Table 1 foods-11-01612-t001:** Changes in whiteness of surimi gels with difference content EGCG during F-T cycles.

Color	EGCG Content (%)	F-T Cycles			
		0	2	4	6
L*	0	72.80 ± 0.40 ^Aa^	71.30 ± 0.13 ^Ba^	70.90 ± 0.07 ^Ba^	70.2 ± 0.29 ^Ca^
	2	71.00 ± 0.59 ^Ab^	70.10 ± 0.11 ^Bb^	69.40 ± 0.47 ^BCb^	69.0 ± 0.12 ^Cb^
	4	70.80 ± 0.11 ^Ab^	69.20 ± 0.36 ^Bc^	68.90 ± 0.14 ^Bc^	68.1 ± 0.25 ^Cbc^
	6	70.40 ± 0.38 ^Ab^	69.20 ± 0.40 ^Bc^	68.60 ± 0.47 ^BCc^	68.0 ± 0.43 ^Cc^
a*	0	−1.60 ± 0.08 ^Ac^	−1.70 ± 0.03 ^Ac^	−1.60 ± 0.01 ^Ad^	−1.70 ± 0.10 ^Ac^
	2	−0.50 ± 0.05 ^Ab^	−0.70 ± 0.04 ^Bb^	−0.80 ± 0.05 ^Bc^	−0.60 ± 0.10 ^ABb^
	4	−0.30 ± 0.04 ^Aa^	−0.60 ± 0.02 ^Bab^	−0.20 ± 0.06 ^Ab^	−0.30 ± 0.08 ^Aa^
	6	−0.30 ± 0.08 ^Ba^	−0.50 ± 0.09 ^Ca^	0.10 ± 0.03 ^Aa^	−0.20 ± 0.04 ^Ba^
b*	0	4.60 ± 0.09 ^Aa^	3.50 ± 0.16 ^Ca^	3.60 ± 0.21 ^Ca^	4.00 ± 0.26 ^Ba^
	2	2.40 ± 0.01 ^Ab^	2.30 ± 0.13 ^ABb^	2.30 ± 0.46 ^ABb^	2.20 ± 0.25 ^Bb^
	4	2.50 ± 0.08 ^Ab^	2.10 ± 0.03 ^Bc^	2.00 ± 0.31 ^BCbc^	1.70 ± 0.18 ^Cc^
	6	2.50 ± 0.09 ^Ab^	1.80 ± 0.06 ^Bd^	1.80 ± 0.08 ^Bc^	1.70 ± 0.13 ^Bc^
W	0	72.40 ± 0.40 ^Aa^	71.00 ± 0.04 ^Ba^	70.70 ± 0.09 ^Ba^	69.9 ± 0.30 ^Ca^
	2	70.90 ± 0.60 ^Ab^	70.00 ± 0.11 ^Bb^	69.30 ± 0.50 ^BCb^	69.0 ± 0.11 ^Cab^
	4	70.30 ± 0.40 ^Ab^	69.10 ± 0.36 ^Bc^	68.90 ± 0.10 ^Bc^	68.0 ± 0.23 ^Cbc^
	6	70.70 ± 0.10 ^Ab^	69.20 ± 0.40 ^Bc^	68.50 ± 0.50 ^BCc^	68.0 ± 0.43 ^Cc^

Uppercase letters indicate significant difference (*p* < 0.05) between different F-T cycle, lowercase letters indicate the difference between gels with different EGCG content (*p* < 0.05), and the values are expressed as mean ± SD. L*: lightness, a*: redness-greenness, b*: yellowness-blueness, and W: whiteness.

## Data Availability

Not applicable.

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
