# Peer review of "Assessing the Gel Quality and Storage Properties of Hypophythalmalmichthys molitrix Surimi Gel Prepared with Epigallocatechin Gallate Subject to Multiple Freeze-Thaw Cycles"

_foods, 2022, doi:10.3390/foods11111612_

Round 1
Reviewer 1 Report
the manuscript authored by Zhihang Tian et al. is aimed to describe properties of a novel formulation of EGCG-surimi gel.
The manuscript is interesting and full of relevant data and experimental paradigms. I just have minor comments:
-Figure 1 and 2, should be more clear. I suggest authors to increase the size of single pictures. Also, the figure legend can be reported only once.
-In fig 3 and 4 , statistical analysis should be included in the figures.
-The legend description should include also statistical analysis and should be more detailed.
Reviewer 2 Report
1. What is the novelty of the presented study ? Authors did not stress that.
2. English needs to be improved
3. Statistical analysis: If the experiment was repeted three times it would be more proper to present results as LS means and SEM
4. Section 3.3 line 223: whitness is the moste intuitive indication for consumers to identify food quality. Every food?
5. Section 3.3 Color. Lines 231-232 Besides quinones ... it is rather primay reason for color change.
5. Section 3.4.1 lines: 253-257. Lack of proper citation. In manuscript there is no results supporting that conclusions.
6. 3.4.2 Lines 270-271. Is there a linear correclation batween EGCG concentration and TVB-N? Please elaborate on
7.Section 3.4. TBA Line 288 EGCG treatment grooups insted of group.
8. Section 3.6.2 Lines 259-361 please reformulate. All water was gone ..." the moisture of the surimi gels was lost".
9.Section 3.7 Lines 376-378 Please reformulate.
10. Generraly discussion is scarce.
